# How to Understand Personalized Medicine in Atopic Dermatitis Nowadays?

**DOI:** 10.3390/ijms24087557

**Published:** 2023-04-20

**Authors:** Alicja Mesjasz, Karol Kołkowski, Andreas Wollenberg, Magdalena Trzeciak

**Affiliations:** 1Dermatological Students Scientific Association, Department of Dermatology, Venereology and Allergology, Faculty of Medicine, Medical University of Gdansk, 80-214 Gdansk, Poland; 2Department of Dermatology, Venereology and Allergology, University Hospital, Ludwig Maximilian University, Frauenlobstr. 9-11, 80337 Munich, Germany; 3Department of Dermatology, Free University Brussels, University Hospital Brussels, Bd de la Plaine 2, 1050 Brussels, Belgium; 4Department of Dermatology, Venereology and Allergology, Faculty of Medicine, Medical University of Gdansk, 80-214 Gdansk, Poland

**Keywords:** atopic dermatitis, monoclonal antibodies, JAK inhibitors

## Abstract

Atopic dermatitis (AD) is a heterogeneous disease in terms of its phenotypical, barrier, and immunological presentation. Emerging therapies are undoubtedly contributing to a new chapter in the treatment of AD, bringing an excellent possibility of individualization, and thereby creating a tailored approach. The two most promising substance groups are biological drugs (dupilumab, tralokinumab, lebrikizumab, nemolizumab) and Janus kinase inhibitors (JAKis) (baricitinib, upadacitinib, and abrocitinib). The vision that certain well-defined phenotypes and endotypes, as well as personal preferences, may guide the future treatment of AD is both tempting and appealing, but not yet reality. The accessibility of new drugs such as biologics and small molecules has opened up the discussion regarding personalized medicine, referring to the complex nature of AD as well as the experiences from clinical trials and real-world evidence. We have now reached the point of creating new strategies and AD treatment goals by increasing the amount of new information concerning the efficacy and safety of new drugs. This article has reviewed the novel treatment options for AD in the light of the heterogeneity of this disease and proposes a broader vision on the strategy of personalized treatment of AD.

## 1. Introduction

Atopic dermatitis (AD) is a common, chronic, inflammatory skin disease affecting approximately 20% of children and 2–8% of adults worldwide [1,2]. Both the prevalence and the severity of symptoms vary among populations and ages. For instance, among children between 6 and 7 years old, the prevalence of AD ranges from 0.2% in China to 24.6% in Columbia [3]. The most characteristic features of AD are pruritus and dryness of the skin with periodic remissions and exacerbations/flares [4]. Clinically, apart from dry skin, the elementary symptom is erythematous inflammatory skin lesions with eczematous morphology, while the chronic phase of the disease involves thickening (lichenification) and exfoliation of the epidermis [4]. Usually, the bends of the elbows and knees, the face, and the neck are affected. However, in some cases, AD may also have an erythrodermic course [4]. Understanding the complex heterogeneous pathogenesis of AD should lead to improvement of the individualized therapy presented to each patient. Therefore, our aim was to review the current possibility of personalizing the treatment of AD using biologic drugs and Janus kinase inhibitors (JAKi).

## 2. Discussion

### 2.1. Heterogeneity of AD

AD is a heterogenic disease not only in terms of ethnicity but also age. Some differences are visible in the clinical picture of AD depending on the age of a patient, while others are observed depending on the age of onset of the disease [4].

Firstly, unique endotypes are influenced the exposome developing the numerous phenotypes of eczema [5,6,7]. The exposome is defined as the measure of all the exposures that an individual has had during their lifetime and how these exposures relate to the individual’s well-being. Perfluoroalkyl and polyfluoroalkyl substances (PFASs), pollution, and maternal nutrition have been shown to be the most important exposomal factors in infancy and childhood [8]. Later on in adolescence, climate, infection, and rural and urban environments are the main players, while in adulthood, lifestyle factors and stress predominate [8]. The genetic and epigenetic signature (endotype) of each individual case leads to the development of complex molecular mechanisms. The extrinsic endotype is more common (~80%), while the intrinsic endotype occurs only in a minority of people (~20%) [5,6,7,9]. The extrinsic endotype is characterized by increased serum IgE levels, a familial predisposition to atopy, and a disruption of the skin barrier, resulting in increased transepidermal water loss (TEWL) [7]. The intrinsic endotype has a similar clinical presentation, but the serum IgE levels are normal, a greater metal hypersensitivity is observed, and it mainly affects females [7]. Importantly, in both endotypes the role of T-helper 2 lymphocytes (Th2) seems to be crucial, although, in intrinsic AD the activation of Th1 and Th17 subset is more important [10]. Moreover, the concentration of Th1 and Th17 cytokines is positively correlated with the scoring of atopic dermatitis (SCORAD) in intrinsic AD [10]. Despite several studies, endotypes cannot currently be defined in a certain and precise way, which makes them impossible to use when choosing therapies for patients.

The phenotypic distinction between AD phenotypes seems to be relevant and practical from the clinician’s perspective. Ethnicity and age are two major factors, which distinguish specific molecular patterns characteristic to each group (European/American AD, Asian AD, African/American AD, pediatric AD, and adult AD) [5,6,7]. When considering immunological differences between Asian AD and European/American AD, the first is characterized by a bigger activation of the Th17 axis [7,10,11]. In contrast to Asian AD, cytotoxic Th1 is thought to be an important player in the chronic phases of European/American AD [7,10,11]. In African/American AD, on the other hand, responsibility is skewed towards the Th2/Th22 subsets, while Th1/Th17 do not seem to be pathogenic [7]. Similarly to African/American AD, Th2-mediated activity has been linked to the atopic march and is crucial in pediatric AD, while Th1 has not been shown to be as important in this group as it is in adults [12,13]. The patient’s age as well as the disease chronicity should therefore be a crucial element when considering a therapeutic approach. Th2 is the most important immunological factor contributing to inflammation in children with AD; the levels of Th2 cytokines and related chemokines (e.g., IL-13, OX40 ligand, and thymic stromal lymphopoietin (TSLP) receptor) were similar or greater in lesional and nonlesional pediatric AD skin compared with those in adults [7]. Moreover, IL-9/Th9 has been found to be significantly higher in children than adults [7]. Concomitantly, defects in lipid metabolism in the skin cells contribute to defects in the epidermal barrier to the greatest extent in children [14]. In contrast to Asian AD, the barrier proteins loricrin (LOR) and filaggrin (FLG) are significantly downregulated in European/Americans AD [15]. In contrast, Asian AD exhibits epidermal alterations including increased hyperplastic (determined by thickness and Ki67 count), parakeratotic, and focal hypogranulosis [15]. African/American AD displays a reduction in loricrin (LOR) expression, but not FLG [15].

Importantly, the most commonly used diagnostic criteria for AD (Hanifin and Rajka criteria) have been developed based primarily on the pediatric population [16]. In the case of adult-onset AD, the diagnosis is clinical and often requires excluding similar dermatoses, especially cutaneous T-cell lymphomas [16]. Therefore, additional tests such as the patch tests, prick tests, skin biopsies, or blood tests, are usually necessary to rule out other diseases and should be performed [16,17]. The most frequent clinical presentations of AD in adults are flexural dermatitis with involvement of the face and neck, chronic hand eczema, and less often, nodular prurigo [16].

### 2.2. T2-Type Immunological Driver for AD Pathogenesis and Comorbidities

Skin barrier disruption, genetic predispositions, environmental factors, skin microbiota dysbiosis, as well as an altered immune response are the five main factors involved in the pathogenesis of AD [1,2]. The FLG mutation is a strong genetic predisposing factor for AD, and is less prevalent in the intrinsic subtype of AD compared with the extrinsic one [18]. In terms of skin barrier disruption, AD is characterized by an increase in TEWL and epidermal permeability, an increase in stratum corneum (SC) pH, a decrease in the expression of structural proteins of the SC, including filaggrin, and SC lipid imbalances [18]. The complex pathophysiology of itch involves the role of well-known cytokines such as Il-4, Il-13, and Il-31. Together with IL-33, TSLP, and histamine, they trigger the sensation of pruritus either directly on the sensory neurons or by modulating the neural transmission, thereby contributing to chronic itch [19]. Recent interest in the neuropathic character of atopic pruritus may result in an upcoming change in therapeutic strategies [20]. Skewed immunity in the Th2-produced cytokines, such as Il-4 and Il-13, is responsible for the pathogenesis of AD across all endotypes [18]. Patients with AD have strong dysbiosis and a reduction in microbial diversity in relation to the extent of Staphylococcus aureus colonization [1,2]. Moreover, the pathogenic role of Malassezia spp. should also be considered, as it may promote the secretion of proinflammatory cytokines such as IL-4, IL-13, and IL-17 [1,2]. In addition, environmental factors, such as greater exposure to allergens, pollution, infection, cosmetics, and harsh detergents, in addition to the duration of breastfeeding as an infant, should be taken into account [21]. We have a wider knowledge on the increased co-occurrence of AD with some cardiovascular, metabolic, neurological, autoimmune, and psychiatric diseases, as documented in several observational studies [22]. Multiple explanations for each disease’s higher co-occurrence with AD exist, beginning with shared risk factors and ending with complex genetic and immunological mechanisms [22]. In this setting, there is an even greater need for efficient drugs to decrease inflammation.

### 2.3. Stratification of AD Patients

However, despite all of the meticulous research on the complex pathogenesis of AD, we still lack the ability to assign patients to subgroups which would allow their treatment to be personalized [23]. The tools that may lead us to fulfill this task are biomarkers—defined by the European Medicines Agency (EMA) as a biological molecules found in blood, other body fluids, or tissues that can be used to follow body processes and diseases in humans and animals [23]. Different types of biomarkers exist—we may distinguish two groups: descriptive (describing the state of a disease but not involved in its pathogenesis) and predictive (involved in the pathogenesis or specific to one disease) [24]. Unfortunately, despite several potential biomarkers having been investigated, currently none have been validated and accepted for use in the clinical management of AD [23,25]. One of the most promising biomarkers appears to be the thymus and activation-regulated chemokine/C-C motif ligand 17 (TARC/CCL17), which correlates with AD severity in children and adults and has been used in Japan in clinical practice since 2008 [23]. In a recent study, elevated levels of TARC/CCL17 in 2-month-old infants have been shown to be significantly correlated with an increased risk of developing AD within first 2 years of life [26]. A two-round Delphi survey performed by BIOMAP (biomarkers in AD and psoriasis) members has highlighted the seven most important features of future biomarkers for AD and psoriasis: reliability, clinical validity, relevance, positive predictive value, independent validation, measuring therapeutic response, and disease progression [27]. A study of 147 serum markers in 193 AD patients revealed four patient clusters with distinct serological markers [28]. High levels of pulmonary and activation-regulated chemokines (PARC), tissue inhibitors of metalloproteinases 1 (MP-1), and soluble CD14 were associated with a higher disease severity and body surface area (BSA) [28]. Further research is needed in order to discover a biomarker fulfilling the mentioned features. The most important, in our opinion, seem to be measuring AD activity and the positive predictive value. By knowing the exact activity of AD, we would be able to determine the appropriate time to intensify therapy. On the other hand, knowing that a flare will manifest in several upcoming days, we would be able to administer stronger treatment and thereby reduce or even omit the exacerbation of the clinical symptoms of the disease. For now, the wide use of biomarkers in clinical practice is a tempting vision; however, it is not yet reality.

### 2.4. Novel Medications for AD

Some possibilities arise from the introduction of novel drugs, namely biologics and small molecule inhibitors. The mechanisms of action of these drugs are illustrated in Figure 1.

### 2.5. Biologic Drugs in the Treatment of AD

Biologic drugs are mainly monoclonal antibodies (mAB). The mechanism of action of mABs is to specifically target extracellular (surface or soluble) antigens, in comparison to small molecule inhibitors, which are bioavailable through the oral and cutaneous routes and target intracellular molecules.

Limited data on the biological half-life of dupilumab exists—the range was from 11.7 to 20.5 days when it was administered to monkeys, bearing in mind that a difference between species exist [29]. Different pharmacokinetic properties result from the possibility of binding to IL-4Rα in humans, but not in monkeys [29]. Therefore, in humans parallel linear and non-linear pathways of elimination of dupilumab exist [30]. The first (linear, non-saturable proteolytic pathway) is primarily active when dupilumab is present in a higher serum concentration, and the latter (non-linear saturable IL-4R α target-mediated elimination) is primarily active when the concentration decreases [30]. Importantly, the last steady-state dupilumab concentration decreases below the limit of detection after 6–7 weeks when administered in the regimen of 300 mg every four weeks, and after up to 13 weeks for the standard regimen of 300 mg every two weeks and 300 mg once a week, administered by injection [30]. The half-life of tralokinumab in humans is approximately 22 days [31]. The mentioned drugs are mostly metabolized in a slower manner. In order to achieve their maximal concentration, 3 to 8 days are required [31,32].

The first registered and meticulously researched, fully human mAB licensed for the treatment of AD was dupilumab, which blocks IL-4Rα—a shared receptor unit for interleukin-4 (IL-4) and interleukin-13 (IL-13) [33]. Dupilumab, and biologics in general, have a significantly longer duration of action than JAKi; therefore, patients do not have to take these agents every day as is obligatory with JAKi. In the study by Brunner et al., proteomic results obtained from blood and transcriptomic skin biopsy data were compared and significant differences were found in lesional and non-lesional tissues between pediatric patients with early-onset AD and adult patients with long-standing AD [34,35]. Defects in the epidermal differentiation complex were only observed in adults with AD, possibly as a result of a chronic immune abnormality that is not present in early-onset disease [34,35]. Interestingly, intervening in immunological pathology at an early stage may allow for the reversal of changes and prevention of their consolidation. Dupilumab may prevent the onset of new or worsening pre-existing allergic conditions in adolescents and adults; consequently, it may have the potential to alter the progression of the atopic march [36]. Moreover, a possibility of a disease-modifying effect also exists, as was seen during the follow-up of the studies (five dupilumab half-lives), and no rebound in allergic events was observed [36]. As the treatment benefit of dupilumab appeared greater for younger patients—aged under 18 years and those with an early onset of AD at less than 2 years of age—it may be theorized that the mentioned effect could also apply in children, in addition to adolescents and adults [36]. This thesis may be backed up by the recent report of a patient with a history of pistachio allergy, who received the allergen during an open food challenge three months after finishing dupilumab therapy. A good tolerance of pistachio was noticed [37]. While treatment with dupilumab may be considered a nonallergen-specific immunotherapy approach for the management of food allergies in the future, further research needs to be carried out as the available data are still limited. The safety profile has been assessed over a period of 172 weeks and outstanding clinical results in relieving symptoms that were previously significant and decreased quality of life have been shown in numerous clinical trials [36,38,39]. Dupilumab is regarded as a safe medication, with some sources suggesting that routine laboratory monitoring is unnecessary during dupilumab therapy [40]. Currently, dupilumab is registered for children older than 6 years and adults in Europe, the United States, and several other countries. Moreover, the FDA has announced that this drug is to be registered in children aged 6 months to 5 years old, who suffer from moderate-to-severe AD [41]. In a real-life observation for 204 weeks, dupilumab was continuously effective for adults with moderate-to-severe AD, as was previously observed in studies conducted over a shorter period [42]. Interestingly, a clinical trial observed tapering of dupilumab doses (300 mg every 4 and every 6–8 weeks), compared with a standard dosage regimen (300 mg every 2 weeks) [43]. Patients were selected for tapering when the disease activity was controlled: eczema area and severity index (EASI) ≤ 7, indicating mild disease activity or less for at least 6 months [43]. In the chosen subgroups, despite a reduction in the dose and lower levels of dupilumab in the patient’s sera in the tapered groups, AD was controlled (EASI ≤  7). The median EASI at the endpoint of the study in the groups with a dosing regimen of 300 mg of dupilumab every 4 and 6/8 weeks was found to be 1.5 and 2.9, respectively [43]. In addition, disease severity biomarker levels (pulmonary and activation-regulated chemokine (PARC/CCL18) and thymus and activation-regulated chemokine (TARC/CCL17)) remained low [43]. No significant differences across all groups in the serum concentration of these biomarkers was been found [43]. However, an important issue of drug blocking—anti-dupilumab antibodies—was not researched in the discussed study [43]. The results of SOLO-CONTINUE are also worth analyzing. In this trial, patients were randomly assigned to groups receiving dupilumab weekly, every 2 weeks, and less frequently—every 4 and 8 weeks [44]. A dose-dependent response was found as the proportion of patients achieving an EASI-75 at week 36 was significantly higher in those receiving dupilumab every week and every 2 weeks (71.6%) than every 4 weeks ((58.3%); nominal *p* < 0.05) or every 8 weeks ((54.9%); nominal *p* = 0.01) [44]. Furthermore, a lower incidence of treatment-emergent anti-dupilumab antibodies was found to be associated with more frequent dosage regimens (11.7% in the every-8-weeks group versus 1.2% in the weekly group) [44]. Therefore, according to SOLO-CONTINUE, less frequent administration of dupilumab (every 4 weeks or every 8 weeks) resulted in reduced efficacy, no safety advantages, and numerically higher treatment-emergent anti-drug antibody incidences [44]. On the other hand, in the TRAVERSE and extension of the TRAVERSE studies, which were performed on patients with asthma, anti-dupilumab antibodies did not have any meaningful effect on efficacy and safety of the treatment [45,46]. Consequently, questions have arisen—Should we focus on the economic capacity by tapering the doses, but instead risking drug resistance? Or maybe anti-dupilumab antibodies did not interfere with dupilumab therapy in patients with AD? Conjunctivitis and blepharitis, which were transient in most cases, were the most common adverse effects that occurred during therapy with this agent [25,47].

IL-13 is another of the Th2-profile cytokines; therefore, blocking it was theorized to bring relief for AD [48]. The central role of IL-13 seems to be supported by the results of trials on lebrikizumab and tralokinumab—agents blocking this cytokine [49,50,51].

ECZTRA1 and ECZTRA2 have proven tralokinumab—a fully human mAB against IL-13—to be safe to use and well tolerated for up to 52 weeks of treatment [51]. Importantly, a recent study has shown that a combination of tralokinumab and TCS may be used efficiently in patients whose disease was not controlled with cyclosporin A, or had contradictions preventing the use of this drug (results of ECZTRA 7) [52]. A total of 17.5% of people receiving the 300 mg dose and 21.4% of those receiving the 150 mg dose met the primary endpoint of 0 (“clear”) or 1 (“almost clear”), according to the investigator’s global assessment (IGA); this was compared to 4.3% with placebo [53]. As a comparison, in a trial with dupilumab conducted on adolescents, this result was achieved for 24.4% of people receiving a 200 mg or 300 mg dose (depending on their body weight) every 2 weeks, 17.9% for the dose of 300 mg every 4 weeks, and 2.4% for placebo [54]. In the Measure Up2 trial, the vIGA-AD response (defined as a vIGA-AD score of 0/1 with ≥2 grades of reduction from baseline) at week 16 was 39% for the 15 mg upadacitinib dose, 52% for the 30 mg upadacitinib dose, and 5% for placebo [55]. Lebrikizumab proved to be a safe and effective drug during 2b [25]. However, when administering tralokinumab and lebrikizumab in the future, we must bear in mind the predisposition to conjunctivitis; fortunately, most of the cases where this was experienced were mild and transient [56,57]. Up to week 16, the incidence of conjunctivitis was 8.4% for dupilumab, 2.6% for lebrikizumab, and in the meta-analysis summarizing the results of 2285 adults with AD, 7.5% for tralokinumab [49,58,59]. Treatment of this conjunctivitis is possible with topical corticosteroid or topical calcineurin inhibitor preparations [60].

Nemolizumab is a mAB that targets interleukin-31 (IL-31) receptor α (IL-31Rα) located on the dorsal root ganglia near the cell bodies of cutaneous sensory neurons [33]. Results suggest that IL-31 is responsible for a pruritogenic mechanism unique to AD [57]. These theoretical assumptions have been confirmed by second-phase RCTs, which concluded that nemolizumab not only improves pruritus in patients, but reduces inflammation and skin symptoms, while having an acceptable safety profile [61,62]. The most common adverse effects were nasopharyngitis and upper respiratory tract infection and only a few common adverse events leading to discontinuation of the therapy have been observed [61,62]. Third-phase RCTs are currently ongoing.

### 2.6. Janus Kinases Inhibitors in the Treatment of AD

Small molecules have a broader spectrum of activity than biological drugs because they can block multiple cytokine pathways. Among their diverse molecular targets are phosphodiesterase 4 (PDE4), neurokinin 1 receptor (NK-1R), histamine H4 receptor (H4R), kappa opioid receptor (KOR), and dihomo-gamma linolenic acid (DGLA) [63]. Among JAKis, it is plausible to distinguish those that are more and less targeted. Furthermore, the selectivity of these small molecules differs between members of the group (Figure 1).

JAKis have a rapid onset of action. First significant itch reduction starts on day 1 with abrocitinib, and on day 2 with baricitinib and upadacitinib [64,65,66]. The half-life of JAKis is measured in hours and stands at 2.8–5.4 h for abrocitinib, 4 h for upadacitinib, and 12.5 h for baricitinib [67,68]. Cytokine pathways are targeted by JAKis, either by oral or topical administration [66]. JAKis may be a viable option for short-term treatment during flare-ups when rapid and visible results are required.

#### 2.6.1. JAK STAT Pathway

The human JAK family consists of four Janus kinases (JAK1-3 and TYK2), whereas the human signal transducers and activators (STATs) family consists of STAT1, STAT2, STAT3, STAT4, STAT5A, STAT5B, and STAT6 [69]. The cytokines, interferons, and growth factors bind to the tyrosine-kinase-associated receptors (RTK), causing the dimerization and subsequent activation of JAK, which later phosphorylates STAT. (Figure 1) STAT dimerizes and is translocated to the nucleus leading to the transcription of targeted genes followed by the release of various cytokines, growth factors, and other molecules [69]. The epigenetic influence of the JAK/STAT pathway has a critical role in the pathogenesis of several immune-mediated diseases including AD [69].

A large amount of data has been gathered from the use of small molecule inhibitors in rheumatology; however, rheumatoid arthritis and AD are diseases with two distinct age groups [70]. Most adverse events of JAKis are mild to moderate in severity [71,72,73]. A higher risk of malignancy is a concern with JAKis, as the suppression of the immune system might reduce anti-tumor surveillance [74]. Although relatively rare, venous thromboembolism (VTE) may occur [74]. The exposure time within short-term clinical trials is relatively limited and long-term studies are needed to reliably assess the real-life risk [74].

According to the EMA, JAKis should not be administered to patients aged 65 and older, those with major cardiovascular risk factors, those who smoke or have smoked for an extended period of time, and those at an increased risk of cancer if no suitable treatment alternatives are available [74].

#### 2.6.2. Baricitinib

Baricitinib is a drug that was approved in 2020 by the EMA for the treatment of adults with moderate to severe AD; 2 mg and 4 mg daily dosages are available [75]. This drug is currently being evaluated for use in children and adolescents aged 2 to 17 years [76]. The BREEZE-AD3 study, designed for adults taking 4 mg and 2 mg of baricitinib for 68 weeks, demonstrated sustained long-term efficacy; however, the long-term safety and efficacy profile of dupilumab is undoubtedly better established [73,77,78]. According to the National Institute for Health and Care Excellence (NICE) guidelines, in the future, baricitinib may be introduced before dupilumab [72,77,79]. The majority of mild to moderate adverse events experienced were upper respiratory tract infections and headaches [72]. An increased risk of infections such as herpes zoster, herpes simplex, and eczema herpeticum is possible, though the incidence was still relatively low [72].

#### 2.6.3. Upadacitinib

In 2021, abrocitinib and upadacitinib became oral JAKis approved by the European Medicines Agency (EMA) for the treatment of moderate-to-severe AD; this was followed by the approval of abrocitinib by the U.S. Food and Drug Administration (FDA) at the beginning of 2022 [80,81]. JAK1 inhibitors block crucial cytokines engaged in the pathogenesis of AD such as Il-4, Il-13, and Il-31, while minimizing the risk of cytopenia caused by JAK2 inhibition [82]. By blocking limited cytokine axes, treatment with JAK1 inhibitors may be safer.

Upadacitinib is a drug registered for once-daily use by the FDA and EMA for patients over 12 years old with moderate-to-severe AD [83,84]. Tablets of 15 mg and 30 mg are available [83,84]. In the Heads Up trial, adults with moderate-to-severe AD received either 30 mg of upadacitinib daily and 300 mg of dupilumab weekly for 24 weeks [58]. A higher proportion of patients treated with upadacitinib achieved EASI75 at week 2 (43.7%) compared with those treated with dupilumab (17.4%) [58]. Additionally, patients receiving upadacitinib experienced significantly higher declines in the mean worst pruritus numerical rating scale (NRS) than those receiving dupilumab, with this tendency starting in week 1 and continuing through week 16 [58]. Upper respiratory tract infections and acne were the adverse effects occurred most frequently with upadacitinib [71]. Serious infections were uncommon, though they occurred more frequently with upadacitinib than in the placebo group [71].

#### 2.6.4. Abrocitinib

Abrocitinib is a drug registered for adults with moderate-to-severe AD in adults in the EU. Doses of 50 mg, 100 mg, and 150 mg are available as tablets to be taken daily. In JADE COMPARE, 200 mg of abrocitinib was superior in terms of treating itch than 300 mg of dupilumab via injection on the checkpoint date after week 16 [85]. In the 16-week study, abrocitinib outperformed dupilumab in difficult-to-treat areas, including the head and neck [86]. The 200 mg dose as monotherapy was the best at achieving disease control, and 200 mg reduced to 100 mg was an acceptable compromise in case of dose-related side effects [87]. Treatment discontinuation after achieving desirable results was an ineffective strategy [87]. The ultimate therapeutic option for flares was 200 mg of abrocitinib with either topical corticosteroids (TCS) or calcineurin inhibitors [87]. Abrocitinib is a generally well-tolerated drug and nausea, headache, and acne were the most common adverse events [73]. Herpes zoster, herpes simplex, and eczema herpeticum were the most commonly experienced adverse events of special interest [73].

### 2.7. Patients’ Treatment Expectations

In order to achieve a truly patient-tailored approach, one should not underestimate patients’ needs and personal preferences and invest some time in expectation management. A tailored approach should have an impact on improving adherence to the therapy, satisfaction from the course of the treatment, as well as the therapy outcomes [88,89,90]. Noncompliance has a detrimental impact on the efficacy of the therapy [91,92,93,94].

The most crucial attribute for patients with AD is skin clearance at week 16, followed by an early reduction in pruritus [92]. Most patients are ready to accept adverse events for the efficacy of the treatment in general, as well as accepting the mode of administration [89]. A once-daily pill is usually chosen over an injection every 2 weeks as the preferred form of administration [92,95]. The prevalence of needle fear has been well documented in another survey highlighting that only 50% of respondents were willing to opt for an injectable treatment [90]. The number of patients who prefer injections increases with age and is higher among those currently receiving biologics [88]. Patients prefer oral medications over topical treatments, often seeing the latter as “less convenient” and “time consuming to apply”, which has a direct consequence on the adherence of the patients [89,95]. Nevertheless, some patients claim a fear of swallowing pills [89]. In terms of safety profile, both short-term and long-term adverse events are considered by patients when determining the preferred treatment [89]. The annual risk of malignancy development has a higher relevance to patients over the annual risk of serious infections and venous thromboembolism, which are equally feared by them [92]. According to another survey, the consideration of the safety profile varies not just among patients of distinct ages, but also among physicians, with caregivers and adults placing a greater weight on this matter than adolescents [89].

### 2.8. Future Perspectives in the Personalized Treatment of AD

Along with the mentioned, patient-tailored approach of therapy, we should focus on delivering good quality in terms of managing AD patients, as stated by the European Task Force on Atopic Dermatitis (ETFAD) [96]. On the third of the proposed levels of quality of care, easy access to novel therapies (mABs and JAKs) should be preserved for interdisciplinary teams (dermatologists, allergologists, pulmonologists, and pediatricians) managing severe cases of AD [96]. In addition, cardiologists, endocrinologists, and gastroenterologists, among others, should be included in the decision-making process for therapeutic options, as patients may have numerous non-allergic comorbidities to which AD additionally appears to predispose. This strategy may help to minimize drug interactions and select the optimal treatment for multiple diseases. In fact, partnership in the decision-making process, which concerns implementing novel treatments, has been shown to be one of the most important needs of AD patients [97]. However, the therapeutic effectiveness of these drugs strongly depends on the molecular mechanisms appropriate for each patient, which are influenced by a considerable number of factors. Currently, due to a lack of validated biomarkers, patients cannot be precisely assigned to subgroups determining that a certain agent would be the best therapeutic option. Another important aspect is financing novel treatments for patients, as the costs of such therapies are high [98]. However, despite the fact that these therapies are expensive, their efficacy can make them well worth the cost [99]. Novel therapies have the potential to limit long and frequent hospitalizations, psychological stress, and impaired school and work performance due to poor management, as well as the high costs of topical treatments, among other things [99]. Nevertheless, particular limitations in various countries exist as well as distinct procedures for reimbursement; thus, accessibility to new treatment options differs between countries.

Due to distinct immune profiles and comorbidities, several clinical trials engaging solely elderly populations in the same manner as pediatric populations should be introduced in the future. This is particularly important in terms of the safety profile of novel therapies, which is pertinent in elderly groups of patients—for example, monitoring renal functions when using cyclosporine A [25]. In a recent article, an important issue of key parameters that should be used when creating future RCTs to assess drugs for AD treatment has been raised [100]. Due to the abundance of RCTs performed on each medication, a need to analyze parameters of comparative efficacy (for instance EASI, IGA, SCORAD, and others) as well as parameters of safety arises [101,102]. Systematizing comparative data between medications will consequently simplify the decision-making process in terms of selecting the ultimate therapy for each patient [101,102].

Along with phenotypes and endotypes, and their corresponding medications, alongside patients’ personal preferences, the selection of the ultimate treatment should be tailored to the comorbidities and exceptional states of the patient such as pregnancy or the lactational period. Additionally, pregnancy is a contraindication for JAKi therapy [103]. Other mABs and JAKs have not been researched in this specific patient population to the best of our knowledge; therefore, their use in pregnancy should be omitted too [25].

A higher degree of precision in drug action seems to be the key to personalized medicine. The objective would be to classify patients according to their immunological features [104]. However, in such a heterogenic disease as AD, strictly grouping patients with certain endotypes is currently unavailable. The question remains as to whether the possibility of creating such an approach even exists, bearing in mind the complexity of the disease. Biomarkers for both AD and AD treatment would undoubtedly be of great assistance in this regard; however, despite the fact that a great deal of research is being conducted worldwide on this subject, no biomarker that is routinely used in clinical practice has yet been identified [25].

The next important point is how fast/early and how widely to commence treatment to stop the disease and comorbidities, and how long to continue treatment. For example, in the case of dupilumab, it is the only drug with a proven efficacy in asthma and AD; therefore, this treatment may be preferred for patients presenting with a coexistence of these two diseases [105]. Furthermore, dupilumab has the most data supporting its use in elderly populations [105]. Should we also search for clinical or immunological markers to indicate the AD subgroup that may potentially need short-term or long-term treatment?

Due to an early onset of action and good effect sizes (EASI-90 and EASI-100 becoming easily reachable), the small molecule inhibitors may be revolutionary for the short-term treatment of AD, and possibly also as an add-on treatment for exacerbations experienced whilst taking biologics. Currently, the data remain limited to short-term clinical trials and a number of studies are essential to clarify the long-term safety and treatment success of JAKis. So far, these drugs have shown promising effectiveness balanced by an acceptable safety profile in 16-week clinical trials [102].

## 3. Conclusions

In summary, JAKis and biologics appear to provide a future possibility for individualizing AD treatment; however, this is not yet established in reality. A need for additional research on the comparative safety and efficacy of the drugs exists, especially in patients with other comorbidities and in special circumstances, such as older age or pregnancy. Furthermore, aligning patients’ individual preferences with the various features of the drug treatments seems to be important. Nevertheless, the safety of the drug treatment for both physicians and patients should remain the most important feature. As of today, the largest experience is with dupilumab treatment (long-term and in real life), and this drug may remain the drug of choice for several patient groups.

Furthermore, we should understand the personalized approach to AD treatment in terms of the meaning of molecular endotypes including biomarkers, phenotypes including comorbidities, and individual patient’s needs, in reference to the efficacy and safety of the new drugs, taking possible AEs into consideration (Figure 2).

## 4. Materials and Methods

A comprehensive search of the literature using the PubMed electronic database with the search queries “(heterogeneity and atopic dermatitis) OR (pathogenesis and atopic dermatitis)”, “abrocitinib and atopic dermatitis”, “baricitinib and atopic dermatitis”, “upadactinib and atopic dermatitis”, “tofacitinib and atopic dermatitis”, “ruxolitinib and “atopic dermatitis”, “delgocitinib and atopic dermatitis”, “dupilumab and atopic dermatitis”, “lebrikizumab and atopic dermatitis”, and “tralokinumab and atopic dermatitis” was performed in the first week of March 2022. The search period was from the inception of the database to 6 March 2022. Further research using the query “personalized and atopic dermatitis” was performed in the second week of March 2022, from the inception of the database to 13 March 2022. After the initial search, titles and abstracts were screened against the inclusion and exclusion criteria. Based on the title and abstract analysis, we included articles concerning the role of biological drugs affecting cytokine profiles and JAKis in atopic dermatitis. At this step, we excluded records not related to the topic, non-English manuscripts, personal opinions, and duplicates. The remaining records were qualified as eligible for full-text reading. After reading the full manuscripts, some were excluded (as they were not relevant and did not provide information concerning the drugs, pathways, and new drug impact on atopic dermatitis mentioned earlier). Critical reviews were taken into consideration. Additionally, the site “clinicaltrials.gov” was searched for three-phase randomized controlled trials (RCTs) of the abovementioned drugs in the treatment of atopic dermatitis. Finally, additional relevant, eligible records identified through a search of the references were included where information on the effect of a personalized approach and novel insights into atopic dermatitis treatment were found. The potential limitation of this study is that over one-hundred articles on the topic of novel therapy in atopic dermatitis have appeared since the literature was reviewed.

## Figures and Tables

**Figure 1 ijms-24-07557-f001:**
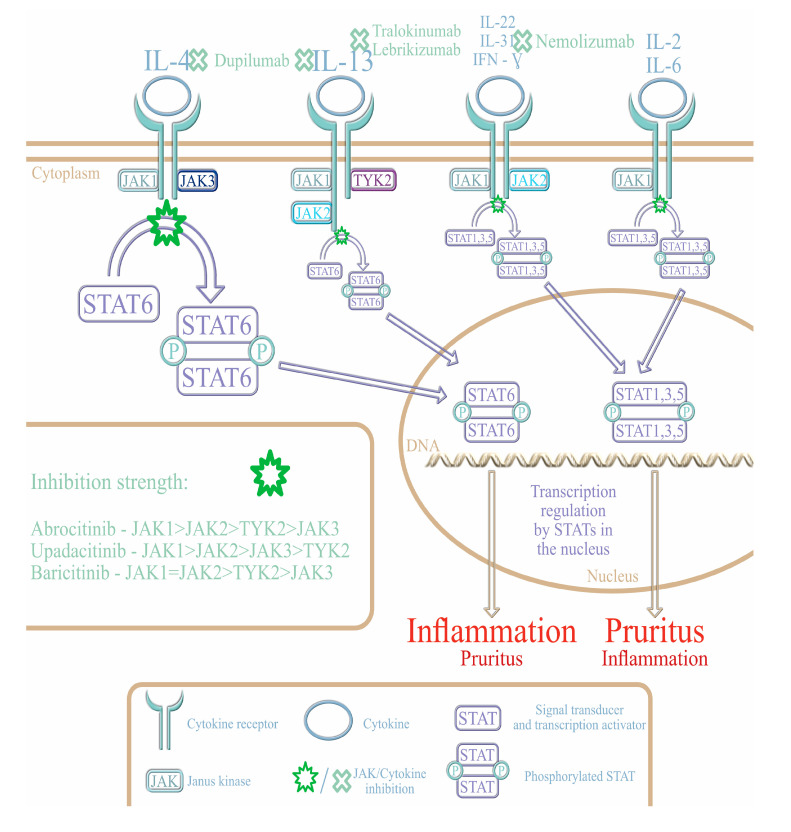
Physiological mechanisms of action of several cytokines important in AD pathophysiology, which activate Janus kinases (JAKs) to phosphorylate signal transducers and transcription activators (STATs), have been illustrated. Two pathways, one activated by interleukin-4 and interleukin-13 mostly promoting inflammation and the second stimulated by other cytokines predominantly promoting pruritus, have been shown. Both biological and small molecule inhibitor drugs may block distinct pathways according to their specific place of action—narrower for biologics (dupilumab, tralokinumab, lebrikizumab, nemolizumab) and wider for JAK inhibitors (JAKi) (abrocitinib, upadacitinib, baricitinib).

**Figure 2 ijms-24-07557-f002:**
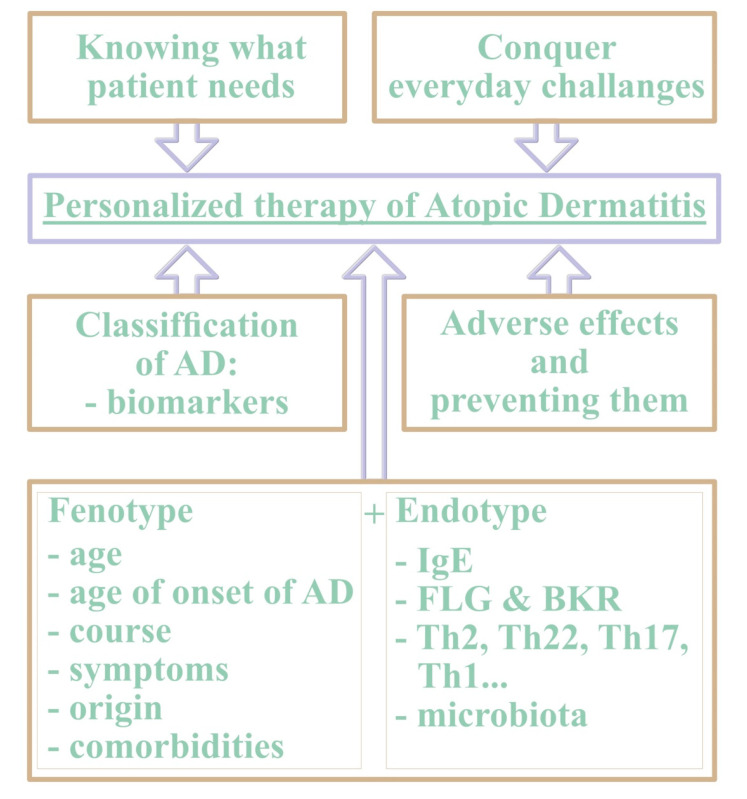
A comprehensive view of the personalized therapy of atopic dermatitis is needed. In this model we propose five main factors to be taken into consideration in order to create a patient-tailored approach: patient’s preferences, real-life challenges and finding reasonable solutions of dealing with them, fast recognition and prevention of adverse effects, phenotypes and endotypes, and lastly, biomarkers, which are not yet available.

## Data Availability

All the data can be found in the PubMed database—https://pubmed.ncbi.nlm.nih.gov/ accessed on 1 February 2023 or using the cited websites.

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
