# Peer review of "How to Understand Personalized Medicine in Atopic Dermatitis Nowadays?"

_ijms, 2023, doi:10.3390/ijms24087557_

Round 1

Reviewer 1 Report

Thank you for the opportunity to read this review summarising recent concepts on the classification and management of atopic dermatitis. While the topic is very interesting, the authors do not provide sufficient information on current and future efforts for achieving a tailored approach in the treatment of this complex disorder. 

Please consider the following issues:

- Introduction: the structure is not clear and some important concepts are not explained at all, such as the role of the exposomes and the clinical phenotypes that are identified in the literature. Moreover there is potential confusion between the concepts of age of onset and disease duration  

- Line 86: “together with many other factors” this is a generic statement that should be explained

- Line 90: “dysbiosis”: a current view on this need to take into account not only the bacterial microbiota but also the fungal one

- Biologics: line 147 “their mechanism of action is narrower” I suggest to revise this concept considering that monoclonal antibodies are large molecules that require parenteral administration and specifically target extracellular (surface or soluble) antigens while small molecules are bioavailable through the oral and cutaneous route and their targets are intracellular molecules

- Line 158: though multiple regimens were investigated in this trial, for clarity please refer to the standard approved regimen of 300 mg q2w

- Line 173-175: “Dupilumab may prevent the onset of new or worsening preexisting allergic conditions in adolescents and adults” is the effect of dupilumab on allergic manifestations limited to when the drug is administered or has a disease modifying effect been demonstrated? Could this be hypothesised in paediatric patients? Moreover, could the presence of allergic comorbidities guide the choice of treatment (preference for dupilumab)?

- Line 191 “AD has been controlled”: this is true for approximately 50% of subject in less frequent dosing groups

- Line 205 “anti-dupilumab antibodies”: are these drug blocking antibodies?

- Line 217: what was the primary endpoint for IgA? How do these results compare to dupilumab and to the JAKis?

- Line 221: conjunctivitis is a major issue for dupilumab: what is the frequency of this adverse event in dupilumab and in anti IL13 mabs?

- Line 227 “but reduces inflammation”: please clarify. Was nemolizumab able to improve skin signs of AD or just reduce inflammatory markers?

- Line 240 “topical” Topical and oral small molecules with targets other than JAKs are part of the current and future treatment armamentarium of AD and should be mentioned by the authors.

- Line 247 “the receptor”: please be more specific on the nomenclature of cytokine receptors that are associated to JAK/STAT signalling

- Lines 315-319 and line 372-376: the meaning here, as in other occurrences, is not clear, please rewrite for clarity

- Line 339 “interdisciplinary”: could comorbitidies be a clinical criteria for the choice of the preferential drug?

Reviewer 2 Report

Personalized approach to AD treatment based on phenotype and endotype would be very interesting, This is an interesting review, This is a great review however there are some questions that should be clarified.

1.       The long term safety is an important question. What do you think about starting these treatments for children and consequences that many years of such therapies could have?  This issue should be discussed with some consideration in more depth.

2.       As reported on p. 8 the high cost of these therapies is an issue to be addressed but given recent works (e.g. DOI: 10.1155/2023/4592087) you could add justified by clinical improvements and the lower use of topical therapies and hospital admissions especially during a pandemic period.

3.       Materials and methods. Some limitations should be discussed considering that from the analyzed period there are more than 100 papers on these topics in the literature.

4.       There are few grammar mistakes. Examples of these:

Pag. 2 line 47 …endytype…“

Round 2

Reviewer 1 Report

Dear Authors,

Thank you for your response. I have sincerely appreciated your efforts in revising this manuscript. Please consider the following minor comments to additionally improve your paper. 

Point 1: Introduction: the structure is not clear and some important concepts are not explained at all, such as the role of the exposomes and the clinical phenotypes that are identified in the literature. Moreover there is potential confusion between the concepts of age of onset and disease duration

Response 1: Thank you for your suggestions. According to them, we have reorganized the introduction and added information on the role of exposome in atopic dermatitis (AD) (lines 57-63). We have also clarified the potential confusion between the concepts of age of onset and disease duration (lines 50-51).

Lines 33-34 “The most characteristic features of AD are pruritus and dryness of the skin with periodical remissions and exacerbations/flares.” The clinical presentation of AD is missing: what are the elementary skin lesions? 

Lines 42-43 “Some differences are visible in the clinical picture of AD depending on age of a patient, while others are observed depending on the age of onset of the disease.” Here, I suggest at least mentioning the clinical phenotypes that have been established in the literature for adult AD; please see: Silvestre Salvador JF, Romero-Pérez D, Encabo-Durán B. Atopic Dermatitis in Adults: A Diagnostic Challenge. J Investig Allergol Clin Immunol. 2017;27(2):78-88. doi:10.18176/jiaci.0138
Additionally, you could briefly report the different endotypes of AD according to ethnicity and age groups (Czarnowicki T, He H, Canter T, Han J, Lefferdink R, Erickson T, Rangel S, Kameyama N, Kim HJ, Pavel AB, Estrada Y, Krueger JG, Paller AS, Guttman-Yassky E. Evolution of pathologic T-cell subsets in patients with atopic dermatitis from infancy to adulthood. J Allergy Clin Immunol. 2020 Jan;145(1):215-228. doi: 10.1016/j.jaci.2019.09.031. Epub 2019 Oct 15. PMID: 31626841; PMCID: PMC6957229.; Czarnowicki T, He H, Krueger JG, Guttman-Yassky E. Atopic dermatitis endotypes and implications for targeted therapeutics. J Allergy Clin Immunol. 2019 Jan;143(1):1-11. doi: 10.1016/j.jaci.2018.10.032. PMID: 30612663.).

Point 6: Line 173-175: "Dupilumab may prevent the onset of new or worsening preexisting allergic conditions in adolescents and adults" is the effect of dupilumab on allergic manifestations limited to when the drug is administered, or has a disease-modifying effect been demonstrated? Could this be hypothesised in paediatric patients? Moreover, could the presence of allergic comorbidities guide the choice of treatment (preference for dupilumab)?

Response 6: Thank you, we have added the information on the modifying effects and the possibility of the modifying effects in paediatric patients (lines 203-208). Moreover, the possible presence of allergic comorbidities has been discussed in the chapter 2.5. Future perspectives in the personalized treatment of AD.

This is correct but I advise a cautious approach to this very important topic. While dupilumab is being increasingly studied as nonallergen specific immunotherapy approach for the management of food allergy (to reduce the risk of anaphylaxis and improve quality of life), long term data on the variation of allergen sensitivity (including following interruption of treatment) is still limited and we must be aware that the beneficial effect of dupilumab might be reversible after treatment interruption. (Albuhairi S, Rachid R. Novel Therapies for Treatment of Food Allergy. Immunol Allergy Clin North Am. 2020;40(1):175-186. doi:10.1016/j.iac.2019.09.007)

Point 7: Line 191 "AD has been controlled": this is true for approximately 50% of subject in less frequent dosing groups

Response 7: Thank you for the suggestion, however, we are not sure to which study you are referring to (line 223). In the study, which we have cited (Spekhorst LS, Bakker D, Drylewicz J, et al. Patient-centered dupilumab dosing regimen leads to successful dose reduction in persistently controlled atopic dermatitis. Allergy. 2022;77(11):3398-3407. doi:10.1111/ALL.15439) in the less frequent dosing groups, the AD has been controlled (defined by EASI ≤  7) in around 90%: "At T2 (dosage Q4W for at least 3 months), 83.3% (n = 25) of the patients in group B, and 86.7% (n = 26) of the patients in group C had controlled AD. No significant differences in EASI score were observed between T1 and T2 in both subgroups (p = .17 and p = .79). At T3, an extended dosing interval of Q6W/Q8W had been applied in group C of which 28 patients (93.3%) had controlled AD, and no significant difference in EASI score was observed compared with T1 (p = .19) (see Table 2 and Figure 2)." Importantly, in the first subgroup, in which the standard dosage regimen has been kept the control of AD has only been achieved in only 40-50% of patients in different time points. We have to bear in mind that these patients had a more severe disease, because they did not meet the criteria to be included in the less frequent dosing groups.

I suggest you also consider data from the SOLO-CONTINUE study. In this study, patients successfully treated in monotherapy with dupilumab 300 mg Q2W or QW for 16 weeks could either continue Q2W/QW administration or receive dupilumab 300 mg Q8W and Q4W. In the latter two cases efficacy was shown to be reduced in a dose-dependent manner. At week 36, EASI-75 response was maintained in 58.3% and 54.9% of patients receiving dupilumab Q4W and Q8W respectively (Worm M, et al. Efficacy and Safety of Multiple Dupilumab Dose Regimens After Initial Successful Treatment in Patients With Atopic Dermatitis: A Randomized Clinical Trial SOLO Continue [published online ahead of print, 2019 Dec 26]. JAMA Dermatol. 2019;156(2):131‐143. http://dx.doi.org/10.1001/jamadermatol.2019.3617).
